# Computer Simulation with a Temperature-Step Frying Approach to Mitigate Acrylamide Formation in French Fries

**DOI:** 10.3390/foods9020200

**Published:** 2020-02-16

**Authors:** Der-Sheng Chan

**Affiliations:** Department of Information Technology, Lee-Ming Institute of Technology, New Taipei City 243, Taiwan; dschan@ms58.hinet.net

**Keywords:** simulation, acrylamide, water content, mathematical model, kinetics

## Abstract

A heat and mass-transfer model coupled with reaction kinetics was developed to simulate frying. Obtaining an accurate mathematical model of the Maillard reaction and the heat and mass transfer is crucial for predicting the transient acrylamide formation, temperature, and water content in French fries. The objective of this study was to mitigate the formation of acrylamide in a potato strip by adopting a temperature step frying approach (TSFA). A considerable increase in the water content and a decrease in the temperature and acrylamide formation were observed in a potato strip fried with the TSFA compared with a potato strip fried without the TSFA process. The acrylamide content in a potato strip when fried using the TSFA decreased considerably to 57% of that in a potato strip fried without using the TSFA. Simulation of the acrylamide distribution in a potato strip revealed that the crust contains the highest amount of acrylamide. The proposed model can be successfully used to obtain high-quality products, mitigate acrylamide formation, and save energy.

## 1. Introduction

Acrylamide is classified by the International Agency for Research on Cancer as a Group 2A probable carcinogen; it is a byproduct of the Maillard reaction that occurs in starchy food processed at high temperatures [1]. Thus, developing appropriate strategies for minimizing the acrylamide content in asparagine-containing baked foods is essential [2,3]. Many official documents published by the European Union (European Food Safety Authority) and United States (Food and Drug Administration) that provide recommendations, guidance, and plans to reduce acrylamide in heated food are available [4,5,6]. Modifying the baking time and temperature can help reduce the amount of acrylamide in baked foods [4]. A useful method for mitigating the formation of acrylamide is to employ a baking process at a low temperature for a long heating time [7,8]. Lowering the temperature during the final cooking stages can lower the amount of acrylamide [9,10]. Acrylamide is found in many high-temperature processed foods, such as French fries, potato chips, baked cereal, and coffee. These foods are a major source of acrylamide [11]. The golden yellow color and delicious flavor, which consumers require, of bakery processed foods is attributable to the heating process. Because color is one of the main qualities of processed foods, pleasant cooking is considered to be complete only when the processed product turns golden yellow. Due to the high reducing sugar content in potatoes, color is an important quality in the foods [12,13,14,15]. The brown color is generally achieved through a combination of the Maillard reaction and caramelization in the later stages of heating when the surface crust reaches a certain high temperature [16]. The desired color of processed foods depends on both the physicochemical characteristics of the raw materials (reducing sugars, and amino acid content) and the processing practices (heating temperature, heating time, and heating mode). A strong correlation exists between the intensity of crust color and acrylamide formation in processed foods [17]. The Maillard reaction is essential for the formation of color and aroma in baked foods; however, it may also cause the formation of a toxic compound, acrylamide, if reducing sugars and free asparagine are present in the raw materials [1,17,18,19]. Processed foods contain crucial precursor-free asparagine and reducing sugars and undergo high-temperature heating for a certain duration, which has an impact on the amount of acrylamide formed [17]. Many studies have observed that acrylamide content increases with heating time and temperature [19,20]. Heating at high temperatures and for long durations enhances acrylamide formation in potato-based foods [12,15,20,21,22]. However, adding acids, such as tartaric and citric acids, can reduce the acrylamide content in processed foods [18,23], mainly because during the later stages of heating, an acidic pH hinders acrylamide formation [18,24]. 

Heat and mass transfer during high-temperature baking may simultaneously cause several complex physicochemical changes and chemical reactions in processed foods, such as water evaporation, crust formation, surface browning, and acrylamide formation [22,25,26,27,28]. Thus, obtaining an accurate mathematical model of the Maillard reaction and heat and mass transfer is crucial for simultaneously predicting the aforementioned changes and reactions to obtain an agreement between the experimental data and estimated curves for acrylamide formation, temperature, and water content. An explicit formulation of the evaporation rate per volume was directly incorporated into a heat and mass balance equation [25,29]. Two-dimensional modeling and simulations for baking were performed to obtain an accurate prediction of acrylamide formation [24,26]. Therefore, understanding transport phenomena completely is essential for controlling and optimizing the frying process and reducing the formation of toxic acrylamide [15,29]. A theoretical approach based on the critical heating temperature and time should be adopted to ensure high frying quality; in this manner, the frying process can be optimized. In other words, frying at the lowest possible heating temperature and for the shortest possible time is necessary for reducing the formation of acrylamide [29]. The formation and degradation of acrylamide were observed in a French fries model [25]. The effects of frying temperature and frying time on the quality of potato-based foods have been extensively studied [20,30,31]. A strong correlation must be obtained between experimental data and the model-estimated values of acrylamide formation, temperature, and water content in baking products. To date, only a few published papers have focused on the correlation between three of the factors: acrylamide formation, temperature and water content profiles in fried food. The temperature can be lowered to decrease acrylamide formation when the French fries have low moisture content in the final stages of frying [20,31]. To date, a temperature-step frying approach (TSFA) using a mathematical model has not received enough attention to mitigate acrylamide formation during frying. The acrylamide formation, temperature, and water content in a strip during frying were investigated to verify the proposed mathematical model based on reaction kinetics and heat and mass transfer. A TSFA was proposed using this mathematical model to mitigate acrylamide formation in French fries while maintaining its quality.

## 2. Methodology

When potato strips are fried in an oil bath, energy is transferred from the hot oil to the surface of the potato strips. The strip temperature depends significantly on the frying temperature. The surface temperature of the potato strips is controlled using the difference between the oil and surface temperatures and the physicochemical properties of the strips. The temperature gradient causes heat transfer from the surface to the inside of a strip via conduction, and water diffuses outward in the opposite direction (i.e., from the inside to the surface of a strip); the water subsequently vaporizes into hot air at a surface temperature >100 °C. However, acrylamide forms on areas of the potato strip where the temperature is ≥120 °C. The relationship between water content and local temperature in a food product has been extensively studied [32]. Therefore, the dynamic responses of the temperature and water content of the strip during frying were analyzed. The initial composition of the strip was 77.2% water, 18.8% carbohydrates, and 2.7% protein [30]. French fries in size of 8.5 × 8.5 × 70 mm were investigated [15,20,21,25]. Numerical model coupling heat and mass transfer was simulated for potato strips with the same size for frying at 170 °C in this study. The result of mathematical modeling are compared with experimental data obtained from the study of Gökmen et al. [20].

### 2.1. Governing Equations and Assumptions 

Frying is a complex phenomenon because chemical reactions simultaneously occur with heat and mass transfer in a food product during high-temperature heating. The oil type has an influence on the product quality of French fries, but the effect of oil type is not considered in the study. Thus, the following assumptions were made:(1)The volume change in the strip during frying is neglected.(2)The effect of oil type on the strip during frying is neglected.(3)The water vaporization rate is negligible when the frying temperature is <103 °C [15].(4)Acrylamide is formed at temperatures ≥120 °C [23].

Based on these assumptions, the heat and mass transfer equations, including the initial and boundary conditions, were developed to study the temperature and water content profiles as well as the distribution of acrylamide formation in the axisymmetrically two-dimensional geometry of the strip. Based on assumptions 2 and 3, two step functions, *S*_w_ and *S_AA_*, were introduced to simulate water evaporation and acrylamide formation, respectively. These functions are defined as follows:(1)$$S_{W}=\left\{ \begin{array}{l} 0\;\mathrm{if}\;T\;<\;376.15\;K\\ 1\;\mathrm{if}\;T\;\geq \;376.15\;K \end{array}\right.$$
(2)$$S_{AA}=\left\{ \begin{array}{l} 0\;\mathrm{if}\;T\;<\;393.15\;K\\ 1\;\mathrm{if}\;T\;\geq \;393.15\;K \end{array}\right.$$

To evaluate the decrease in the acrylamide content in strips by using the TSFA, a step change in frying temperature is a function of the frying time. A temperature step frying parameter, *S*_H_, which is a function of the initial frying time (t_H_), is defined as follows:(3)$$S_{H}=\left\{ \begin{array}{l} 1{\;\mathrm{if}\;t\;<\;t}_{H}\\ f\;\mathrm{if}\;t\;\geq {\;t}_{H} \end{array}\right.$$
where f is a numerical parameter for the step frying temperature. For the 170 °C step frying process, f was set as 0.95489, and the frying temperature was changed from 170 to 150 °C ((170 + 273.15) × 0.95489−273.15) after 3 min and 4 min of frying at 170 °C, respectively. The equation for the rate of water vaporization per unit volume of strips, *R*_evp_ (mol/(m^3^.s)), was defined as the amount of vapor produced at a temperature ≥373.15 K:(4)$$R_{evp}=S_{W}k_{evp}C_{w}$$
where *k*_evp_ is the water evaporation rate constant (1/s), *C*_w_ is the water concentration (mol/m^3^).

Figure 1 presents a side view of a potato strip for studying the frying process. Only a half section was required for simulation due to the symmetry of the strip. The equations of heat and mass transfer with water vaporization have been well developed in the literature [31,32]. In this study, the equations of heat and mass transfer with water vaporization are modified as follows:

#### 2.1.1. Heat Transfer

(5)$$\rho_{e}C_{P,e}\frac{\partial T}{\partial t}=\frac{\partial }{\partial x}(\kappa_{e}\frac{\partial T}{\partial x})+\frac{\partial }{\partial y}(\kappa_{e}\frac{\partial T}{\partial y})-R_{evp}H_{evp}M_{W}$$
where *ρ**_e_* is the effective density of the strip, *C_p,_**_e_* is the effective specific heat, and *k**_e_* is the effective thermal conductivity.

Initial condition: (6)$$T(x,y)=T_{i}\;\mathrm{for}\;0\leq x\leq \frac{W}{2}\;\mathrm{and}\;0\leq y\leq H$$
where $T_{i}$ is the initial strip temperature, W is the width of the strip, and H is the height of the strip.

Boundary conditions: (7)$$\frac{\partial T}{\partial x}=0\;\mathrm{for}\;x=0\;\mathrm{and}\;0\leq y\leq H$$
(8)$$\kappa_{e}\frac{\partial T}{\partial x}=h_{t}(S_{H}*T_{fry}-T)\;\mathrm{for}\;x=\frac{W}{2}\;\mathrm{and}\;0\leq y\leq H$$
(9)$$\kappa_{e}\frac{\partial T}{\partial y}=-h_{t}(S_{H}*T_{fry}-T)\;\mathrm{for}\;y=0\;\mathrm{and}\;0\leq x\leq \frac{W}{2}$$
(10)$$\kappa_{e}\frac{\partial T}{\partial y}=h_{t}(S_{H}*T_{fry}-T)\;\;\mathrm{for}\;y=H\;\mathrm{and}\;0\leq x\leq \frac{W}{2}$$
where $h_{t}$ are the convective heat transfer coefficients on the surface of the strip; $T_{oil}$ is the frying temperature; and $D_{e}$ is the effective diffusivity of water.

#### 2.1.2. Moisture Transfer


(11)$$\frac{\partial C_{w}}{\partial t}=\frac{\partial }{\partial r}(D_{e}\frac{\partial C_{w}}{\partial x})+\frac{\partial }{\partial y}(D_{e}\frac{\partial C_{w}}{\partial y})-R_{evp}$$


Initial condition:(12)$$C_{w}(r,z)=C_{w}{}_{i}\;\mathrm{for}\;0\leq x\leq \frac{W}{2}\;\mathrm{and}\;0\leq y\leq H$$
where $C_{wi}$ is the initial concentration of water in the strip.

Boundary conditions: (13)$$\frac{\partial C_{w}}{\partial x}=0\;\mathrm{for}\;x=0\;\mathrm{and}\;0\leq y\leq H$$
(14)$$-D_{e}\frac{\partial C_{w}}{\partial x}=k_{m}(C_{w}-C_{eq})\;\mathrm{for}\;x=\frac{W}{2}\;\mathrm{and}\;0\leq y\leq H$$
(15)$$-D_{e}\frac{\partial C_{w}}{\partial y}=-k_{m}(C_{w}-C_{eq})\;\mathrm{for}\;y=0\;\mathrm{and}\;0\leq x\leq \frac{W}{2}$$
(16)$$-D_{e}\frac{\partial C_{w}}{\partial y}=k_{m}(C_{w}-C_{eq})\;\mathrm{for}\;y=H\;\mathrm{and}\;0\leq x\leq \frac{W}{2}$$
where $k_{m}$ is the convective mass transfer coefficient and $C_{eq}$ represents the equilibrium water content.

### 2.2. Kinetics of Acrylamide Formation and Degradation

The kinetics of acrylamide formation and degradation have been reported in the literature [25,29,33,34]; they are represented as follows:(17)$$A+B\overset{k_{1}}{\to }AA\overset{k_{2}}{\to }D$$
where A, B, AA, and D denote reducing sugar, asparagine, acrylamide, and degraded products melanoidin, respectively. The acrylamide formation can be described using the first-order reaction with respect to reducing sugar [15,25]. Thus, the reduction rate of reducing sugar and the formation rate of acrylamide are expressed as follows: (18)$$\frac{\partial \;\;C_{A}}{\partial \;t}=-k_{1}\;C_{A}$$
(19)$$\frac{\partial \;C_{AA}}{\partial \;t}=k_{1}\;C_{A}-k_{2}\;C_{AA}$$
where $C_{A}$ is the concentration of reducing sugar and $C_{AA}$ is the concentration of acrylamide.

Initial conditions: (20)$$C_{A}=C_{Ai},\;C_{AA}=C_{D}=0$$
where *k*_1_ and *k*_2_ denote the first-order rate constants, which are functions of temperature and have the following general forms: (21)$$k_{1}=S_{AA}k_{0}\exp\;(\frac{-E_{a,AA}}{R_{g}T})$$
(22)$$k_{2}=S_{AA}k_{0}\exp\;(\frac{-E_{a,D}}{R_{g}T})$$
where *k_0_* is the pre-exponential factor, and *E_a,AA_*, and *E_a,D_* are activation energies for the formation of acrylamide and degraded products, respectively. The symbols of the equations are defined in the nomenclature section.

### 2.3. Material Properties

Many ingredients (water, protein, and carbohydrates) were present in the strip. Water is a good conductor of heat; however, the vaporization of water is an energy-intensive process. Water vaporization reduces the temperature of the strip considerably. In the present work, the effective properties of the strip, including density, heat capacity, and thermal conductivity, were estimated using a mass fraction average mixing rule based on the composition and local temperature [35]. 

#### 2.3.1. Density

The density for each component was defined using Equations (23)–(25), and the effective density of the strip was calculated using the mass fraction average mixing rule (Equation (26)). The temperature is expressed in °C in the following equations [35].
(23)$$\rho_{carb}=1.5991\;\times \;10^{2}\;-\;0.31046T$$
(24)$$\rho_{pro}\;=\;13.299\;\times \;10^{2}\;-\;0.51814T$$
(25)$$\rho_{w}\;=\;9.9718\;\times \;10^{2}\;+\;3.1439\;\times \;10^{-3}T\;-\;3.7574\;\times \;10^{-3}T^{2}$$
(26)$$\rho_{e}=\Sigma \;\rho_{j}\;\omega_{j}$$
where *ρ**_car_**_b_* is the density of carbohydrates, *ρ**_pro_* is the density of protein, *ρ**_w_* is the density of water, *ρ*_j_ is the density of the jth compound, and ω_j_ is the mass fraction of *j*th compound.

#### 2.3.2. Heat Capacity

The heat capacity was defined for each component (Equations (27)–(29)) and then calculated using the mass fraction average mixing rule (Equation (30)). The temperature is expressed in °C in following equations [35]:(27)$$C_{P,carb}\;=\;1.5488\;+\;1.9625\;\times \;10^{-3}T\;-\;5.9399\;\times \;10^{-6}T^{2}$$
(28)$$C_{P,pro}\;=\;2.0082\;+\;1.2089\;\times \;10^{-3}T\;-\;1.3129\;\times \;10^{-6}T^{2}$$
(29)$$C_{P,w}\;=\;4.1762\;+\;9.0862\;\times \;10^{-5}T\;-\;5.4731\;\times \;10^{-6}T^{2}$$
(30)$$C_{P,e}=\Sigma C_{P,j}\;\omega_{j}$$
where *C_p,ca_**_rb_* is the specific heat of carbohydrates, *C_p,_**_pro_* is the specific heat of protein, *C_p,w_* is the specific heat of water, and *C_p,__j_* is the specific heat of the *j*th compound.

#### 2.3.3. Thermal Conductivity

The thermal conductivity can also be calculated using the aforementioned method. The thermal conductivity of each component is expressed as a function of temperature in °C [35]. An effective thermal conductivity, *κ_e_*, can thus be calculated using the average of the component mass fraction.
(31)$$\kappa_{carb}\;=\;2.014\;\times \;10^{-1}\;+\;1.3874\;\times \;10^{-3}T\;-\;4.3312\;\times \;10^{-6}T^{2}$$
(32)$$\kappa_{pro}\;=\;1.788\;\times \;10^{-1}\;-\;1.1958\;\times \;10^{-3}T\;-\;2.7178\;\times \;10^{-6}T^{2}$$
(33)$$\kappa_{w}=5.7109\;\times \;10^{-1}\;+\;1.7625\;\times \;10^{-3}T\;-\;6.7036\;\times \;10^{-6}T^{2}$$
(34)$$\kappa_{e}=\Sigma \;\kappa_{j\;}\;\omega_{j}$$
where *k_ca_**_rb_* is the thermal conductivity of carbohydrates, *k**_pro_* is the thermal conductivity of protein, *k_w_* is the thermal conductivity of water, and *k_j_* is the thermal conductivity of the *j*th compound.

#### 2.3.4. Diffusivity

The effective diffusion coefficient of liquid water was defined as a function of the strip temperature as follows:(35)$$D_{e}(T)=D_{0}\exp\;(\frac{-E_{a,dif}}{R_{g}T})$$
where $D_{0}$ is the pre-exponential factor, and $E_{a,dif}$ is activation energies for the diffusivity of water.

## 3. Results and Discussion

### 3.1. Model Validation

Non-linear partial differential equations were solved using COMSOL Multiphysics 5.1 with the finite element method. Table 1 lists the model parameters used in the simulation. Figure 2 illustrates the profiles of average water content in the strip when it was fried at 170 °C. The average water content of the strip decreased from 77.2% to 32.4 and 25.1% when fried for 6 min and 9 min, respectively. The simulated values of water content were in good agreement with the experimental data at frying temperatures; thus, the validity of the mathematical model was proven.

Figure 3 illustrates the temperature profiles of the strip obtained during frying at 170 °C; two distinct periods can be observed in this figure. In this study and as reported in many studies [20,25] as well, in the early stage of frying (0–1.5 min), the surface temperature increased quickly and then gradually (1.5–9 min). After 1.5 min of frying, the surface temperature increased at an even slower rate, and toward the end of frying, the temperature gradually became constant, mainly because when the temperature of the strip surface became higher than the boiling point of water, a heat insulation layer may have formed for the remaining frying duration. Furthermore, the final surface temperature of the strip when fried at 170 °C for 9 min was 148 °C. After 1.5 min of frying, the core temperature was constant (~103 °C). Figure 3 illustrates that the simulated profiles were consistent with the experimental data.

Gökmen et al. [15,20,25] proposed a good mathematical model and indicated that the model could suitably describe the acrylamide content. Due to only one potato strip being fried at a time, no changes in the oil temperature were observed in their studies. If many potato strips were placed at the same time in a fryer, the oil temperature drops rapidly. If a fryer has high heating capacity, the oil temperature can quickly return to the set temperature, which had less impact on product quality. Conversely, if a fryer has low heating capacity, the oil temperature cannot return to the set temperature, which has a great impact on product quality. Thus, with low heating capacity, the frying time prolongs as well as the acrylamide content may increase. A study reported that acrylamide formation was strongly affected by the temperature and water content of the fried surface [20]. Figure 4 presents the relationship between the simulated response and experimental data of acrylamide content in the strip when frying is performed at 170 °C. The average acrylamide formation in the strip increased with the frying time and a similar trend has been reported in many studies [20,25]. At the end of the frying process, the average acrylamide content in the strip fried at 170 °C for 9 min was 418 µg/kg. In the early stage of frying (0–1.5 min), no acrylamide was formed in the strip. On comparing Figure 4 with Figure 3, it is apparent that the strip temperature was <120 °C in the early stage, which limited acrylamide formation. Figure 4 indicates that the lag time for the acrylamide formation was 1.5 min when frying at 170 °C. After the lag time, acrylamide formation takes place rapidly. Figure 4 shows that the simulated acrylamide content and measured acrylamide content are in good agreement. 

### 3.2. Simulation of Frying with a Temperature-Step Frying Approach

Lowering the thermal input by decreasing the frying temperature or time is a key step for saving energy during baking [28]. Gökmen et al. [15,20] found that combining frying temperature with frying time is a suitable approach for reducing the acrylamide formation of French fries. A TSFA was proposed in this study to optimize frying with the aim of saving energy and reducing acrylamide formation. In the TSFA, frying takes place at a frying temperature of 170 °C for 3 min and for 4 min, followed by frying at a frying temperature of 150 °C, until frying is complete. Figure 5 shows the average water content of the strip during frying with the TSFA for 3 min and 4 min. The average water content of the strip decreased as the frying temperature decreased. The average water content in the end fries was 36.8% and 35.0% when frying with the TSFA at 170 °C for 3 min and 4 min, respectively. The average water content in strips when frying at 170 °C with the TSFA was only 3%–4% higher than that when frying at 170 °C without the TSFA method (Figure 2).

The temperature profiles of the strip are highly dependent on the frying conditions. The average temperatures of the strip fried with the TSFA at 170 °C are shown in Figure 6. The temperature of the strip increased sharply to 103 °C during the early stage of frying (1.5 min). Thereafter, the temperature increased very slowly and gradually became constant at 113 °C that was lower than the minimum temperature required for acrylamide formation. The mitigation of acrylamide formation in a product during heating has been extensively studied [15,17,20]. Temperature program frying has been recommended for reducing the acrylamide formation in French fries [15]. The average acrylamide formation in the strip when fried using the TSFA at 170 °C is plotted in Figure 7. In the early stage (0–1.5 min), the acrylamide formation in a strip fried with the TSFA at 170 °C was negligible, owing to the temperature of the strip being <120 °C. The acrylamide content increased considerably when the frying time was >2 min at 170 °C. In the strips fried with the TSFA at 170 °C for 3 min and for 4 min, approximately 178 and 197 µg/kg of acrylamide were formed, respectively. Figure 4 and Figure 7 indicate that the acrylamide content in the strips fried with the TSFA at 170 °C for 3 min was 57 % (418→178, (418 − 178)/418) of the acrylamide content in the strips fried without the TSFA process. The information obtained from the simulation with the TSFA provides guidelines to alleviate acrylamide formation.

### 3.3. Comparison of Frying with and without the Temperature-Step Frying Approach 

Carrieri [29] presented a two-dimensional profile to simulate the distributions of acrylamide in potatoes. The extent of acrylamide formation in potato samples was strongly related to the frying temperature and time [29]. According to De Bonis and Ruocco [37], the distributions of temperature and moisture concentration in freshly cut vegetables during drying depend on the hot air flow field and the temperature of the vegetable samples [36,37]. The distribution of acrylamide in potato samples was highly dependent on the local temperature distribution [29]. The residual moisture of an ideal French fries should be in a range between 38% and 45% for the industrial product [20]. In this study, the final product moisture was set 40%. An effective criterion for comparing frying with the TSFA and frying without the TSFA for the same frying time was investigated in this study. Two-dimensional distribution curves of local temperature, water content, and acrylamide formation in strips can be obtained using a valid mathematical model. The distributions of water content in the strip fried without the TSFA at 170 °C for 9 min and that fried with the TSFA at 170 °C for 9 min are illustrated in Figure 8. The water content ranging from 40% to 77.2% in the strip was smaller when fried normally than that in the strip when fried with the TSFA. The water content in the strip fried with the TSFA was approximately 27.6% higher than that in the strip fried without the TSFA (19.6%). Generally, the water content strongly depends on the local temperature in the fresh-cut vegetable slices [37]. Simulated temperature distributions in the strip fried with and without the TSFA at 170 °C for 9 min are illustrated in Figure 9. The highest temperature was observed at the outmost corner crust of the strip where the rate of heat transfer was the highest. By contrast, the lowest temperature was observed at the core of the strip in both frying processes. The temperature distribution ranges from 103 to 160 °C when frying without the TSFA at 170 °C for 9 min (Figure 9a). However, when frying with the TSFA at 170 °C for 9 min, the temperature distribution was of a much lower range (103–136 °C); the temperature was mostly <120 °C, which is the minimum temperature required for acrylamide formation. Figure 9 shows that the temperature distributions decreased considerably when the strip was fried with the TSFA. Moreover, frying with the TSFA can effectively save energy.

Figure 10 presents the simulated distributions of acrylamide formation in the strip fried with and without the TSFA at 170 °C for 9 min. For both frying with and without the TSFA, the acrylamide was mostly formed at the outmost corner crust of the strip; this is the area with the highest temperature, as illustrated in Figure 9. The relationship between the local sample temperature and acrylamide formation has been well studied [29]. For the strip fried at 170 °C for 9 min, a large amount of acrylamide was formed, ranging from 0 to 2950 µg/kg. In the strip fried with the TSFA at 170 °C for 9 min, a very low amount of acrylamide was formed, ranging from 0 to 1950 µg/kg, mostly on the outmost crust of the strip. The acrylamide of ideal French fries should be lower than 500–600 µg/kg [4,6]. In this study the final acrylamide content was set at 500 µg/kg. The acrylamide content ranging from 0 to 500 µg/kg in the strip was smaller when fried normally than that in the strip when fried with the TSFA. From the aforementioned results, it can be confirmed that frying with the TSFA reduces acrylamide formation to a large extent. 

## 4. Conclusions

Based on results of a computer simulation, a mathematical model for reaction kinetics and heat and mass transfer was developed for frying processes with and without the TSFA. This model not only provided a good understanding of frying but also accurately predicted the water content, temperature, and acrylamide formation in strips. The water content, temperature, and acrylamide formation in strips fried without the TSFA were observed to be quite different from those in strips fried with the TSFA. However, a two-dimensional contour plot was used to investigate and predict the distributions of acrylamide formed in the strip fried with the TSFA and that fried without the TSFA. In the strip fried with the TSFA, a decrease of 43% (418→178, 178/418) was observed in the acrylamide content from the computer simulation (compared to the acrylamide content of the strip fried without the TSFA) at 170 °C for 9 min. The TSFA can be used to mitigate acrylamide formation in strips and save energy during frying. The mathematical model coupled with the TSFA can help promote the development of optimal frying processes.

## Figures and Tables

**Figure 1 foods-09-00200-f001:**
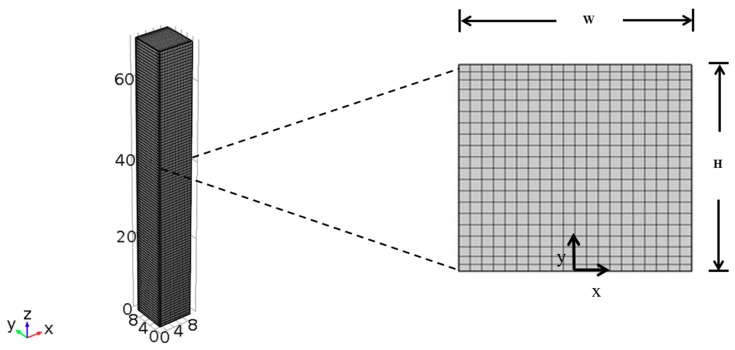
Schematic of the frying process; side view of the strip. *H* is the height of the strip sample; *W* is the width of the strip sample; and x and y are the coordinates.

**Figure 2 foods-09-00200-f002:**
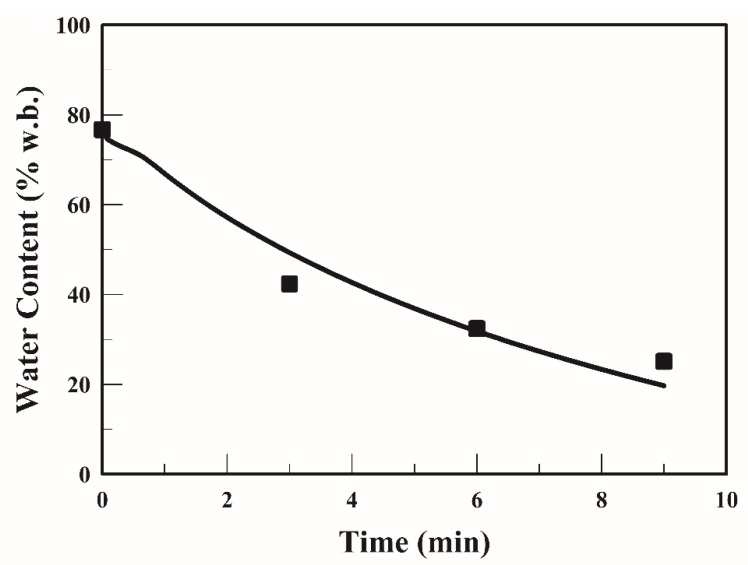
Profiles of the average water content of the strip at 170 °C (■). The symbol represents experimental data [20]; the curve represents simulated values.

**Figure 3 foods-09-00200-f003:**
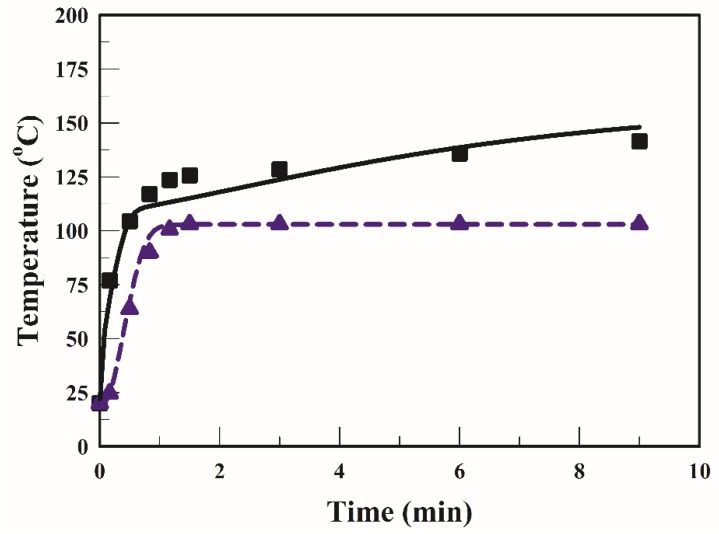
Temperature of the strip at surface (■) and at core (▲) with frying temperature of 170 °C. The symbols ■ and ▲ represent the experimental data obtained [20]; the curves represent the simulated values: surface (solid line) and core (dotted line).

**Figure 4 foods-09-00200-f004:**
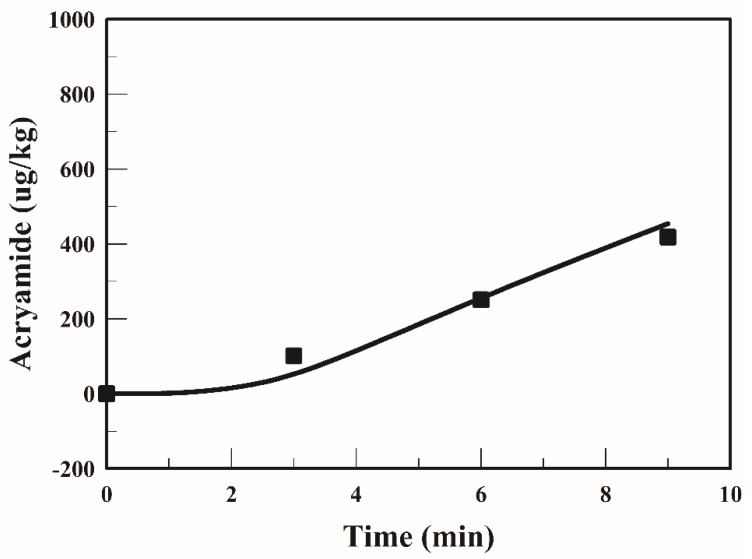
Average acrylamide formation in the strip at 170 °C. The symbol represents the experimental data obtained [20]; the line represents the simulated values.

**Figure 5 foods-09-00200-f005:**
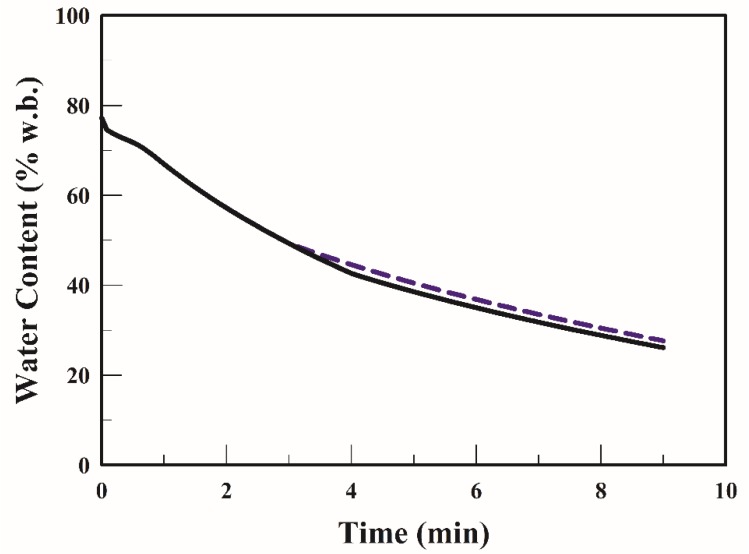
Profiles of the average water content in the strip when fried with the temperature-step frying approach at 170 °C for 3 min (dotted line) and for 4 min °C (solid line).

**Figure 6 foods-09-00200-f006:**
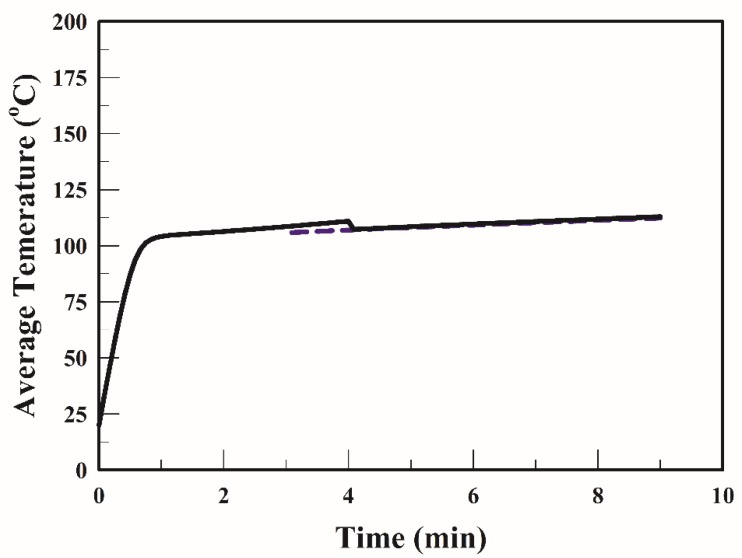
Profile of average temperature of the strip when fried with the temperature-step frying approach at 170 °C for 3 min (dotted line) and for 4 min °C (solid line).

**Figure 7 foods-09-00200-f007:**
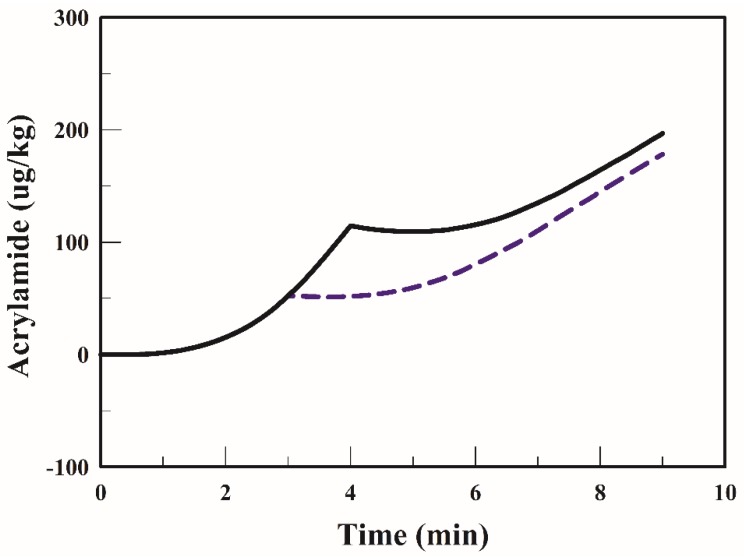
Profiles of acrylamide formation rate in the strip when fried with the temperature-step frying approach at 170 °C for 3 min (dotted line) and for 4 min °C (solid line).

**Figure 8 foods-09-00200-f008:**
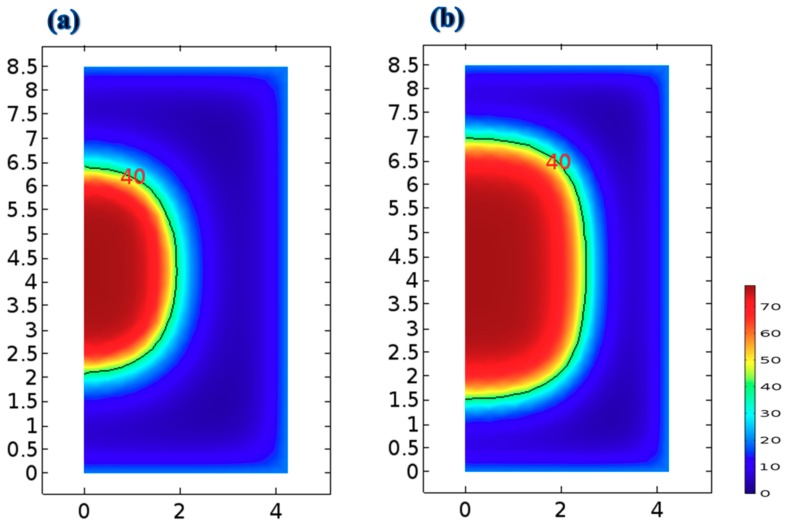
Distributions of the water content in the strip after 9 min of frying: (**a**) without the temperature step frying approach at 170 °C and (**b**) with the temperature step frying approach at 170 °C.

**Figure 9 foods-09-00200-f009:**
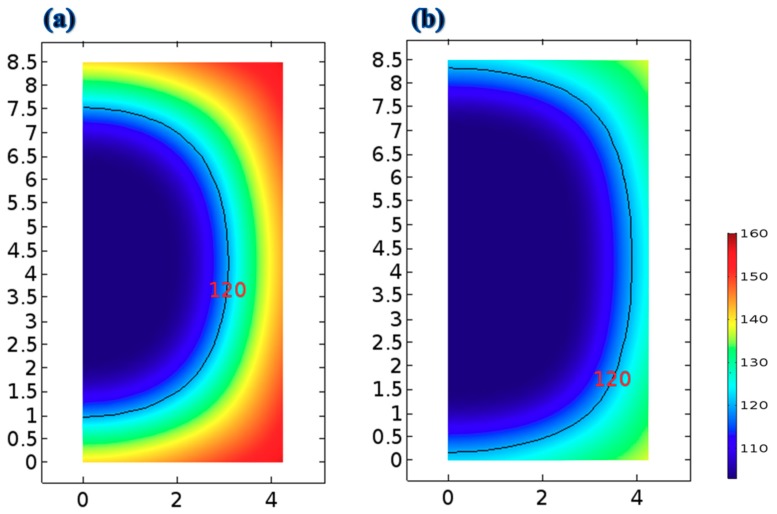
Temperature distributions in the strip after 9 min of frying: (**a**) without the temperature-step frying approach at 170 °C and (**b**) with the temperature-step frying approach at 170 °C.

**Figure 10 foods-09-00200-f010:**
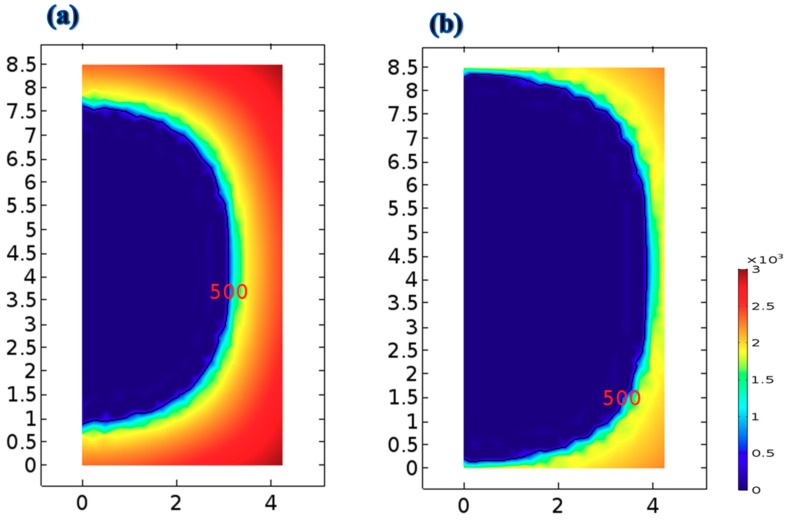
Distributions of acrylamide in the strip after 9 min of frying: (**a**) without the temperature-step frying approach at 170 °C and (**b**) with the temperature-step frying approach at 170 °C.

**Table 1 foods-09-00200-t001:** Input parameter values.

Par. Value		Par. Value		Par. Value		Par. Value	
$C_{Ai}$	6.8 mol/m^3 a^	$E_{a}{}_{,AA}$	44.5 kJ/mol ^e^	$H_{evp}$	2.3 × 10^6^ J/kg ^d^	$k_{0}$	22 1/s ^e^
$C_{eq}$	9265 mol/m^3^ ^e^	$E_{a}{}_{,D}$	21.3 kJ/mol ^e^	$h_{t}$	105 W/(m^2^ K) ^e^	$M_{A}$	0.172 kg/mol ^e^
$C_{wi}$	35,778 mol/m^3 b^	$E_{a}{}_{,dif}$	27.6 kJ/mol ^e^	$k_{evp}$	8.0 × 10^−^^3^ 1/s ^e^	W	8.5 × 10^−^^3^ m ^e^
$D_{o}$	5.0 × 10^−^^6^ m^2^/s ^e^	H	8.5 × 10^−^^3^ m ^a^	$k_{m}$	5.2 × 10^−^^4^ m/s ^e^	$T_{i}$	293.15 K ^s^

Superscripts: ^a^ calculated from [15]; ^b^ calculated from [30]; ^d^ obtained from [36]; ^e^ estimated; ^s^ set; Abbreviation: Par., parameter.

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
