# Peer review of "Computer Simulation with a Temperature-Step Frying Approach to Mitigate Acrylamide Formation in French Fries"

_foods, 2020, doi:10.3390/foods9020200_

Round 1
Reviewer 1 Report
Dear Author The manuscript entitled “Computer Simulation with a Temperature Step Frying Approach to Mitigate Acrylamide Formation in French Fries” used computer to evaluate the effect of different factors that influence to the acrylamide formation and by using TSFA. My suggestion as the following:
Introduction
- All introduction is mention the acrylamide formation in bread and bakery products starting from line 37 until the end of introduction only few sentences about French fried in introduction. Should change the introduction to mention the previous work done on French fried.
- Line 80-83 this sentence should be in the methodology not in introduction.
Methodology
-Did the type of oil effect your hypothesis? - In line 100 the French fries size (8.5 × 8.5 x 70 mm3 ) it is not correct. Many be you mean (8.5 × 8.5 x 70 mm)??
Results
-Line 290: 170 C should change to 170 ºC
Conclusion
-Line 364: It is not clear how you obtain that the TSFA decrease the acrylamide about 61%?
- How the other researchers or company can used TSFA to mitigate the acrylamide formation on the final product?
Author Response
Dear academic editor and reviewer:
We very appreciate your comments and suggestions. The following is our responses to your comments.

Reviewer 2 Report
The paper reports the development of model of a heat and mass transfer coupled with reaction kinetics to simulate frying process. The objective was to reduce formation of acrylamide by application of temperature step frying approach.
The result of mathematical modeling are compared with experimental data, as I understand obtained from literature? It is not clear and should be stated in methodology part. I think that a lot of experimental data are available on this subject, so what criterion was used for data selection? If experimental results were obtained by Author more details concerning experimental conditions should be provide, for example – what oil was used for experiments, what was design of oil bath, how the temperature was controlled, etc. Moreover, the description of methods for determination of chemical composition of strip –water, carbohydrates and proteins (Lines 230-232), and acrylamide formation should be provided in Methodology section.
The system considered for model development is very simple, and only single strip is considered. I doubt that a model based on such a system can correctly describe the process of frying in real conditions? Please discuss this issue.
The effect of TSFA is demonstrated only by simulated data (Fig.5-7) and respective discussion, in my opinion it would be crucial to confirm these results by experiments.
Figure 8-10, The axis labels in the Figures 8-10 should be added.
The author concluded that the strip fried with the TSFA had lower acrylamide content then those fried in constat temperature. It should be clearly state that this conclusion is based on simulated data (?).
Author Response

(The authors gave the same response as above.)

Round 2
Reviewer 1 Report
In my opinion all my suggestion have been answered properly
Reviewer 2 Report
Accept in present form.